# A Review of Digital Eye Strain: Binocular Vision Anomalies, Ocular Surface Changes, and the Need for Objective Assessment

**DOI:** 10.3390/jemr18050039

**Published:** 2025-09-05

**Authors:** Maria João Barata, Pedro Aguiar, Andrzej Grzybowski, André Moreira-Rosário, Carla Lança

**Affiliations:** 1Ophthalmology Department, Unidade Local de Saúde de São José, 1150-199 Lisbon, Portugal; mjsbarata@gmail.com; 2Institute for Research and Advanced Training, University of Evora, 7004-516 Evora, Portugal; 3Department of Therapy and Rehabilitation Sciences, Escola Superior de Tecnologia da Saúde de Lisboa (ESTeSL), Instituto Politécnico de Lisboa, 1990-096 Lisbon, Portugal; 4Comprehensive Health Research Centre (CHRC), Escola Nacional de Saúde Pública, Universidade NOVA de Lisboa, 1600-560 Lisbon, Portugal; pedroaguiar@ensp.unl.pt; 5Department of Health Strategies, National School of Public Health, NOVA University of Lisbon, 1600-560 Lisbon, Portugal; 6Institute for Research in Ophthalmology, Foundation for Ophthalmology Development, 61-701 Poznan, Poland; ae.grzybowski@gmail.com; 7Comprehensive Health Research Centre (CHRC), NOVA Medical School, Faculdade de Ciências Médicas, NMS, FCM, Universidade NOVA de Lisboa, 1169-056 Lisboa, Portugal; 8NOVA Medical School, Faculdade de Ciências Médicas, NMS, FCM, Universidade NOVA de Lisboa, 1169-056 Lisboa, Portugal; 9Division of Science, New York University Abu Dhabi, Saadiyat Marina District, Abu Dhabi P.O. Box 129188, United Arab Emirates

**Keywords:** Digital Eye Strain, binocular vision, convergence, accommodation, dry eye, ocular surface disease

## Abstract

(1) Background: This study investigates the impact of digital device usage on the visual system, with a focus on binocular vision. It also highlights the importance of objective assessment in accurately diagnosing and guiding therapeutic approaches for Digital Eye Strain Syndrome (DESS). (2) Methods: A comprehensive narrative review was conducted to synthesize existing evidence. The methodological quality of observational and case–control studies was assessed using the Newcastle–Ottawa scale, while randomized controlled trials (RCTs) were evaluated using the Cochrane risk-of-bias (RoB 2) tool. (3) Results: Fifteen articles were included in this review, with a predominant focus on binocular vision anomalies, particularly accommodative and vergence dysfunctions, as well as ocular surface anomalies related to DESS. Clinical assessments relied primarily on symptom-based questionnaires, which represent a significant limitation. The included studies were largely observational, with a lack of longitudinal and RCTs. In contrast, research in dry eye disease has been more comprehensive, with multiple RCTs already conducted. (4) Therefore, it is essential to develop validated objective metrics that support accurate clinical diagnosis and guide evidence-based interventions. Conclusions: It remains unclear whether changes in binocular vision are a cause or consequence of DESS. However, prolonged screen time can exacerbate pre-existing binocular vision anomalies due to continuous strain on convergence and accommodation, leading to symptoms. Future research should prioritize prospective longitudinal studies and well-designed RCTs that integrate objective clinical measures to elucidate causal relationships and improve diagnostic and therapeutic frameworks.

## 1. Introduction

Digital Eye Strain Syndrome (DESS) is characterized by the presence of visual symptoms caused by prolonged use of screens [1] impacting both quality of life and productivity [2]. As electronic devices are widespread, people of all ages may be at risk of developing DESS [1]. Individuals with pre-existing eye conditions, such as dry eye disease (DED), allergies, cataracts, glaucoma, and retinal disorders, reported a threefold increase in eye complaints during lockdowns due to prolonged device use [3]. A growing trend has been observed in adults between 18 and 79 years, with most spending 2 to 6 h a day (h/d) using multiple screens [4]. Digital reading differs from traditional paper-based media and is believed to influence the visual system [5]. Screen-based reading seems to be associated with more pronounced subjective ocular symptoms and greater tear film instability when compared to paper-based reading [5]. While oculomotor behavior of eReaders appears largely comparable to reading on paper, greater demands on the visual system are placed by computer screens, as reflected in longer fixation durations [6].

Nowadays, smartphones are the most widely used devices. They are typically used at close distances (25 to 40 cm) and under inadequate lighting conditions, with small text. In university settings, most students rely on electronic study methods, with approximately one-third reporting frequent use of virtual learning [7]. This pattern appears to reflect an increase compared to the pre-pandemic period [8]. Remote work increased, further exacerbating DESS symptoms [9,10]. This review draws on studies published after the COVID-19 pandemic, such as global lockdowns and the significant increase in screen exposure, combined with a reduction in outdoor leisure time, contributed to heightened scientific interest in this topic.

DESS patients report symptoms including headaches, eye irritation, redness, itching, dryness or tearing, light sensitivity, a heavy feeling in the eyelids or forehead, blurred vision, and double vision [11]. These may be classified into accommodative and binocular vision stress or external symptoms linked to dry eye. While often transient, these symptoms can be persistent and frequent, significantly impacting working-age individuals [12]. In a cross-sectional study of video display terminal (VDT) workers, longer screen time (>4 h/d) was associated with increased eye complaints (*n* = 194), particularly among older workers [13]. Additionally, DESS prevalence in VDT workers was 23.4%, compared to 2.9% in the control group (non-VDT workers; <4 h/d), with age (OR = 1.05) and screen time (OR = 1.57) identified as risk factors [13].

Furthermore, dry eye symptoms associated with DESS are primarily triggered or exacerbated by behavioral and environmental factors such as a reduced blink rate and incomplete blinking. This leads to increased exposure of the ocular surface during prolonged screen time [14,15,16,17,18]. However, it remains unclear whether DESS represents a syndromic manifestation of overlapping visual and ocular surface stressors, or if it constitutes a distinct clinical entity.

Healthy binocular vision relies on a balance between convergence and accommodation, which some have hypothesized may be disrupted by excessive screen use [17,18,19,20]. Several studies have investigated the effects of digital device use on accommodation, with mixed results. While some report significant reductions in accommodative function following digital task exposure [e.g., reduced accommodative facility or lag] [15,21], others have found minimal or no measurable impact [18,19]. This inconsistency emphasizes the need for a comprehensive review to synthesize existing evidence and clarify potential factors contributing to these divergent findings. While several prior reviews have addressed DESS [11,12,22,23,24,25,26,27], this narrative review provides a detailed analysis of binocular vision outcomes and highlights the importance of objective evaluation methods to improve diagnosis and treatment efficacy. To our knowledge, no post-COVID-19 review has comprehensively examined the association between DESS and binocular vision anomalies (such as accommodative or vergence dysfunctions). Additionally, it examines ocular surface alterations because of their significant role in the symptomatology of DESS. Importantly, ocular surface disease may mimic or overlap with symptoms caused by binocular/accommodative dysfunctions. This review is guided by three aims: (1) to summarize the reported prevalence of DESS, particularly in the post-COVID-19 period; (2) to examine the relationship between DESS and binocular/accommodative dysfunctions; and (3) to examine the relationship between DESS and ocular surface signs and symptoms of dry eye.

## 2. Materials and Methods

A comprehensive literature search was performed using PubMed and Web of Science databases. The primary search was completed on 2 March 2024 and updated on 16 April 2025. First, a search conducted using the search terms “prevalence” and (“Digital Eye Strain” or “Eye Strain”) identified 29 observational studies, published between 2020 and 2024, that allowed for the assessment of the global distribution of DESS prevalence (Figure 1). These data were analyzed to create a world map using ArcGIS Pro. Secondly, a search was conducted including terms such as “Digital Eye Strain”, “Eye Strain”, “Computer Vision”, “Binocular Vision”, “Convergence”, “Accommodation”, “Dry Eye”, “Tear Film”, and “Ocular Surface Disease”, alongside combinations of these terms using appropriate Boolean operators. The selected terms enabled the inclusion of articles addressing non-strabismic binocular dysfunctions, encompassing accommodative and vergence anomalies such as accommodative insufficiency, accommodative excess, and accommodative infacility, alongside convergence insufficiency, convergence excess, and fusional vergence dysfunction. The references of the selected articles were reviewed to identify any additional relevant studies. This resulted in 15 more articles being included (Figure 1). Within non-strabismic binocular dysfunctions, the selected studies objectively assessed positive relative accommodation (the maximum ability to stimulate accommodation), negative relative accommodation (the maximum ability to relax accommodation), accommodative facility (commonly assessed with ±2.00 D flippers, based on the number of cycles completed within a set time), near point of accommodation (measures in all included studies using a RAF ruler), near point of convergence (measures in all included studies a RAF ruler), as well as both positive and negative fusional vergence (using prism bars and in one study using a synoptophore).

This review included both observational and interventional studies that investigated DESS in relation to at least one of the following parameters: binocular vision outcomes and ocular surface findings. Studies were eligible for inclusion if they met the following criteria: (1) involved human participants, (2) assessed DESS either through symptoms or clinical signs, and (3) evaluated one or more of the relevant visual parameters. Exclusion criteria were as follows: (1) case reports, editorials, and review articles; (2) studies published in languages other than English, Spanish, or French; and (3) studies that did not address DESS in relation to the specified visual outcomes. To refine the selection process, filters were applied to prioritize randomized controlled trials (RCTs), controlled clinical trials, and observational studies. Articles were initially screened based on their titles and abstracts. The language restrictions (English, Spanish, or French) allowed a detailed and accurate evaluation of the selected studies, ensuring a rigorous and reliable assessment of the methodology and findings. However, language restrictions may have resulted in the exclusion of relevant publications in other languages.

The quality of non-randomized studies (*n* = 13) was evaluated using the Newcastle–Ottawa scale, which assesses selection, comparability, and outcomes across a scoring system of 0 to 8 stars, categorizing studies as “low”, “moderate”, or “high” quality [28]. For RCTs (*n* = 2), the Cochrane risk-of-bias tool (RoB 2) was used to evaluate bias across five domains: randomization, intervention deviations, missing data, outcome measurement, and reported results, rating the risk of bias as “low”, “some concerns”, or “high” [29].

## 3. Results and Discussion

### 3.1. Prevalence of DESS

A wide range of DESS prevalence rates was reported in the literature, varying from 42% to 95%, with the lowest rates observed in children under 18 and healthcare professionals, and the highest among university students and digital workers (Appendix A).

DESS prevalence appears comparable between developed and developing countries. However, in developed nations like the USA, China, Spain, Italy, and South Africa, greater access to digital devices often results in higher screen time (Figure 2). In contrast, lower- and middle-income countries may face poorer working conditions and lower awareness, which can increase DESS risk. It is important to emphasize that the reported data reflects only countries where studies have been conducted and published. The absence of data from other countries does not imply that DESS is not present in those countries.

### 3.2. Symptoms of DESS

The etiology of DESS is multifactorial and remains under investigation. Currently, there is no gold standard for its objective assessment, with diagnosis primarily relying on patient-reported symptoms. Subjective tools such as questionnaires are commonly used to identify DESS, based on self-reported experiences of digital device users. The Computer Vision Syndrome Questionnaire (CVS-Q) was introduced in 2015 when computers were the predominant digital device [30]. It includes items similar to those in the Convergence Insufficiency Symptom Survey (CISS) and specifically address symptoms such as headaches, blurred vision, double vision, eye pain, and heavy eyelids (Figure 3). It also includes items similar to those in the Dry Eye Questionnaire version 5 (DEQ-5), particularly regarding symptoms like burning in the eyes, itching, and foreign body sensation (Figure 3).

For effective diagnosis and treatment of DESS, there is a need for further research into its causes. Accurate DESS diagnosis requires objective measures, as symptom-based assessments often overlap with conditions like convergence insufficiency. It remains unclear whether DESS is distinct from convergence insufficiency or an exacerbated form due to prolonged screen use, as both share symptoms such as double vision, fatigue, headaches, and blurred vision (Figure 4).

### 3.3. Binocular Vision Anomalies

The role of binocular vision in DESS has gained attention since 2011, following Rosenfield’s exploration of ocular causes, treatments, and binocular vision issues such as convergence and accommodation disorders [12]. Convergence insufficiency is characterized by near exophoria, a remote near point of convergence, and reduced positive fusional vergence [31]. Near tasks increase the demand on convergence and accommodation, an effect that is further exacerbated by reduced stereo cues during reading, which may make it more difficult for the eyes to maintain fusion [31]. A previous study found that visual symptoms reported by individuals were associated not with screen-time duration but with underlying refractive, accommodative, or binocular disorders, which require proper diagnosis [32].

In this review, we included ten articles on binocular vision [15,19,21,32,33,34,35,36,37,38], comprising nine cohort studies and one case–control study, that were assessed as being of moderate quality, with the Newcastle–Ottawa scale (scores ranging from 5 to 6; Appendix A).
Accommodative Dysfunction and Vergence Dysfunction

A pilot study in young adults (aged 18–23) with normal vision and no dry eye or significant accommodative or binocular vision disorders found a significant reduction in accommodative facility after 60 min of smartphone use (Table 1) [15]. Similarly, in a cohort study conducted amongst university hospital employees in France, fusional amplitude (tested using a synoptophore) and binocular accommodative facility were reduced in employees with more than 5 h/d of screen use (*n* = 28), compared with those using screens less than 5 h/d (*n* = 24; Aufret et al. [36], Table 1). Near point of accommodation (NPA) did not differ between the groups [36]. The refractive error, as assessed by the autorefractor, was also compared with the participants’ spectacles prescriptions [36]. However, no statistically significant differences were found between the refractive status or the adequacy of the correction in the exposed and control groups [36]. We believe that including an assessment of fusional vergence at near in free space, along with the integration of widely used questionnaires such as the CVS-Q, would have been valuable. The small sample size also limited the study, emphasizing the need for larger-scale cohort investigations to better clarify the pathophysiology of DESS. Another limitation to note was the absence of a dry eye evaluation, which may represent a potential confounder. Furthermore, symptomatology was evaluated using the Ocular Discomfort Questionnaire rather than the CVS-Q, limiting comparability with other studies.

In another study involving 93 participants [35], a positive correlation was observed between the CVS-Q scores and positive relative accommodation (Table 1), suggesting that higher symptom levels may be linked to abnormal values in positive relative accommodation. By contrast, no association was observed between the CVS-Q scores and either negative relative accommodation or accommodation amplitude [35]. Additionally, after a short reading period of 20 min, accommodative changes were positively correlated to factors such as time to first fixation, total fixation duration, total reading duration, and reading speed [35]. Although this study was among the most comprehensive in terms of binocular vision assessment, its interpretation was limited by the absence of a dry eye evaluation, an important confounder related to near-work-related symptoms. Expanding the sample size, including a wider age range and accounting for participants’ educational background, would improve the study’s applicability and generalizability. In occupational settings, these findings may have implications for visual performance, productivity, and overall quality of life. This hypothesis warrants further longitudinal studies to better understand the long-term visual consequences of sustained screen exposure.

Similarly, a cross-sectional study in Nepal found that children aged 7–17 years who used digital devices for more than 3 h/d had significantly reduced accommodative amplitudes and facility, supporting the association between extensive screen use and impaired accommodative function [21]. One limitation of the study was the absence of subjective symptom assessment via questionnaires, which restricts the ability to link clinical findings with self-reported symptoms [21]. Nevertheless, the results emphasize that prolonged use of digital devices may have a detrimental impact on binocular vision function in children [21].

To further explore potential interventions for accommodative dysfunction in DESS, a cross-sectional study investigated whether binocular and accommodative factors could explain the effect of low-powered convex lenses (+0.50D, +0.75D, +1.25D) in adults aged 20–40 diagnosed with DESS using the CVS-Q [19]. The study also examined whether these lenses were associated with improved reading performance, measured by the Wilkins Rate of Reading Test (WRRT). While many participants performed better with +0.50D and +0.75D lenses, no consistent association was found between binocular vision anomalies and either reading performance or symptom severity. A small subset with esophoria (5%) demonstrated improved WRRT performance [19]. The study was limited by the absence of detailed ophthalmic assessments and objective measurements of digital screen exposure. Overall, the findings suggest that binocular and accommodative anomalies are unlikely to be primary contributors to DESS. However, the observed improvement in participants with esophoria emphasizes the need for further research into the potential role of heterophoria in DESS.

Building on the need to further investigate the role of heterophoria and other binocular factors in DESS, a subsequent study examined digital device users aged 20 to 34 years, who reported an average of 5 h/d [39]. The authors found that 22.5% exhibited binocular vision dysfunctions, with convergence excess being the most prevalent. However, no statistically significant differences in symptom scores were observed between individuals with and without such dysfunctions. A notable limitation was the absence of a gold standard for diagnosing binocular vision disorders, specifically the lack of a standardized assessment protocol incorporating heterophoria, visual acuity, refractive error, and a full evaluation of binocular and accommodative functions, such as fusional vergences, NPC, NPA, and accommodative facility. This absence limited the ability to make meaningful comparisons across studies and populations and was not unique to this study but a recurring issue across studies included in this review.

In a younger population from Spain, vergence disorders were more prevalent amongst children with severe DESS compared with those who had mild DESS [38]. Statistically significant differences were observed in the near point convergence (break and recovery points) and distance negative fusional vergence (break and recovery points). However, the cross-sectional nature of this study precludes determining causality between DESS and convergence insufficiency [39]. In addition to accommodative findings, Maharjan et al., reported reduced positive fusional vergence (near and distance) in those using devices more than 3 h/d compared to those using devices for less than 3 h/d [21]. Again, symptom questionnaires were absent, reducing clinical interpretability. Supporting these observations in adults, Auffret et al. found that a group with an average screen exposure of 6.7 h exhibited a reduction in fusional vergence and binocular accommodative facility, whereas no such changes were observed in the control group, whose average screen time was 2.1 h [40]. While binocular vision anomalies have been observed in individuals with DESS, evidence identifying the pre-existence of conditions such as convergence insufficiency and accommodative insufficiency (AI) in these individuals is lacking. In addition, orthoptic treatment for convergence insufficiency is known to be effective in correcting vergence and accommodation abnormalities and reducing asthenopic complaints [41]. Thus, if convergence insufficiency is present, treatment could theoretically resolve DESS symptoms as measured by subjective questionnaires such as the CVS-Q. Nonetheless, this hypothesis warrants further investigation to be substantiated.

A key limitation identified in the existing literature (Table 1) is the use of diverse questionnaires, which may have contributed to conflicting findings. In this review, four studies utilized the CVS-Q the most widely adopted and linguistically validated questionnaire, while the remaining six employed alternative questionnaires, such as the Symptom Questionnaire for Visual Dysfunctions (SQVD), the Computer Vision Symptom Scale 17 (CVSS17), and other non-standardized instruments. Additionally, inconsistencies in exposure duration, ranging from studies based on self-reported screen time to those involving standardized tasks of 1 h or as short as 20 min, make direct comparisons difficult due to the lack of a consistent temporal framework. Participant age also varied considerably across studies, which is a critical factor known to influence results, and the sample sizes were small in five of the studies and are not representative of the population, with only one study adopting a longitudinal design. Additionally, the types of digital tasks performed, and the methods used to assess accommodative parameters differed widely (NPC, fusional vergence and accommodative facility tests). Therefore, although the included studies were generally of moderate quality, these findings highlight the need for more rigorous research employing robust protocols that adequately control for confounding variables. At present, it remains unclear whether changes in binocular vision are a cause or consequence of DESS. Lack of longitudinal studies RCTs limits the understanding of DESS and its impact on binocular vision. Existing studies often suffer from design flaws, small sample sizes, non-representative populations, poorly defined data collection methods, lack of standardized protocols, and an absence of validated questionnaires, which hinder the replication and generalization of findings.

### 3.4. Ocular Surface Anomalies

As a secondary aim, we also reviewed ocular surface anomalies, particularly to explore whether symptoms attributed to DESS might be confounded by the presence of binocular vision anomalies. Ocular surface changes have been associated with DESS, including disruptions in the tear film and dry eye disease (DED), due to altered blinking patterns during digital device use [42]. Six studies were analyzed, four of which were cross-sectional [4,15,16,42] and two RCTs [17,18]. Notably, one of these studies also incorporated assessments of binocular vision, allowing for a more integrated evaluation of potential overlapping mechanisms [15]. In the quality assessment using the NOS scale, one study was classified as low quality and five as moderate (Appendix A). When using the RoB2 tool, one manuscript was classified as “some concerns” and one RCT was classified as “low risk bias” (Appendix A).
Display-Related Factors Influencing Ocular Surface

To explore the impact of screen time and the potential for behavioral mitigation, a study involving Chinese university students (*n* = 120) examined the effects of different screen types on digital asthenopia. Participants read for 2 h at 40 cm under four conditions: light and Organic Light-Emitting Diode (OLED), light and electronic ink (eINK), dark and OLED, and dark and eINK (Table 2). Post exposure, the OLED group showed reduced tear meniscus height and increased conjunctival hyperemia, while the eINK group demonstrated greater tear film stability. Further research is required to explore the impact of brightness, flicker frequency, and short-wavelength blue light (450–470 nm) on these changes [17]. Although this was an RCT study, a key limitation was the lack of specification regarding the method of randomization and the blinding of groups, as the display technologies were visually distinguishable to participants [19].

Another study examined the effects of screen polarization on digital asthenopia. The participants were divided into groups using linearly and circularly polarized screens for a two-hour reading test under varying lighting conditions. The group using linearly polarized smartphones showed a significant reduction in Non-Invasive Break-Up Time and increased Ocular Surface Disease Index scores, indicating greater fatigue. Conversely, the circularly polarized screen group showed no significant changes, suggesting that circular polarization, resembling natural light, may reduce eye strain [18]. The authors acknowledged that the main limitation of the study was the lack of adjustment for statistical multiplicity. Such adjustments enhance the likelihood of identifying statistically significant results when multiple comparisons or outcomes are tested within a study. Without proper corrections (e.g., Bonferroni or Holm adjustments), this can lead to false positives, which may appear statistically significant but lack clinical relevance or reproducibility. In the absence of this adjustment, the analysis appears less robust, with the findings showing statistical significance but lacking clinical relevance, thereby necessitating further investigation to determine their clinical importance.
Blinking

Reduced blink rate and incomplete blinking have been identified as key factors contributing to DESS, particularly during prolonged smartphone use [15]. A study involving 12 young adults reading a novel on smartphones for 1 h found that prolonged smartphone use was associated with DESS; however, the study’s small sample size was a major limitation [15]. Symptoms such as fatigue and eye discomfort were associated with reduced binocular accommodation and higher frequency of incomplete blinking, particularly among frequent and prolonged smartphone users, with younger individuals being particularly affected [15]. The findings highlight the need for further research on automated blink detection, tear film and ocular surface integrity, accommodative function, and working distance. A deeper understanding of these effects could aid in developing effective management strategies and clinical interventions for smartphone-related symptoms.

Subsequently, a study of health professionals in Mangalore, India, reported that 24.2% of participants had dry eye [42]. Among these, 47.7% reported that closing their eyes provided the greatest relief. Other commonly used relief measures included taking breaks (32.5%), moving around (15.4%), sleeping (8.4%), and washing the face with water (11%). Interestingly, only 7.8% considered blinking to be effective. The study also highlighted differences in blinking frequency during computer use, emphasizing the importance of blinking patterns in managing DESS symptoms and offering practical insights for those experiencing ocular discomfort [42]. In 2023, an Australian research group published the first study examining the impact of smartphone use on the ocular surface in school-aged children [14]. The study found that just 1 h of smartphone gaming in children with healthy eyes led to a marked decline in ocular comfort and a significant reduction in blink rate, accompanied by prolonged eye-opening intervals [14]. These results suggest a potential long-term risk of ocular surface disease and dry eye associated with excessive smartphone use in children. The authors emphasized the need for further research to better understand these effects and to inform preventive strategies in this increasingly exposed population [14]. In another study, the same research group reported that blinking in children was not influenced by age or regular use of digital devices and was comparable to that observed in adults [43]. One explanation for the discrepancy between the two studies lies in methodological differences. The experimental study exposed participants to a controlled one-hour smartphone gaming session, during which blinking behavior and ocular symptoms were measured in real time using eye-tracking technology [14]. In contrast, the cross-sectional study assessed participants under normal conversational conditions without experimental stimulus [43]. In addition, the instruments used to measure symptoms and ocular surface parameters varied, namely the questionnaires and method of collecting objective measurements. These variations in study design, measurement tools, and environmental context likely contributed to the differing outcomes. Furthermore, both studies involved relatively small sample sizes (*n* = 36 and *n* = 45), which influence the reliability and generalizability of the findings. The research group suggested that future studies should explore blink amplitude, both complete and incomplete blinks, as an important biomarker of ocular surface health, along with its associations with tear function and digital device use [43]. **Collectively, both studies [14,43] indicate that context and task intensity play a critical role in** how digital screen exposure affects ocular health in children. While routine device use may not inherently disrupt ocular surface homeostasis, prolonged and uninterrupted screen activities, such as gaming, can acutely alter blink patterns and lead to discomfort, potentially contributing to ocular surface stress if such behaviors are repeated frequently.
Dry Eye Disease

DED has been extensively studied in relation to DESS, with evidence showing that prolonged digital device use adversely affects the ocular surface, reducing tear BUT and compromising tear film integrity, thereby increasing the risk of developing DED [33,44]. A study in México found that VDT workers with moderate to severe screen exposure showed more pronounced signs of DESS and DED than those with mild exposure. These findings stress the need to limit screen exposure (below 5.96 ± 2.5 h/d or 31.69 ± 14.1 h/week) and implement preventive measures to mitigate the development of DESS and DED [16]. Recommended strategies included consciously blinking during screen exposure, limiting prolonged reading on small screens, optimizing workspace ergonomics, maintaining appropriate viewing distance and screen positioning, using anti-reflective protective spectacles, and taking regular breaks by looking into the distance [16].

Expanding on the pathophysiological understanding of DED, a study on the etiological subtypes of DED identified production deficiency and evaporative subtypes, both linked to demographic and lifestyle factors [45]. These subtypes of DED were associated with risk factors such as older age, elevated stress levels, and poor self-perceived health [45]. Aqueous-deficient DED was more common in women and in those with reduced sleep, while evaporative DED was linked to East and South Asian ethnicity, contact lens wear, and increased screen time [45]. According to Wolffsohn et al., these findings provide valuable guidance for future research by helping to identify groups at risk and by highlighting modifiable factors relevant to DED prevention and treatment [45]. **Building on this evidence,** a study conducted at a medical college hospital explored the association between DESS and DED among healthcare professionals [42]. Most participants reported concurrent use of smartphones and laptops, with 56.1% engaging in simultaneous use and 46.3% reporting 3–5 h/d of device use. Upon evaluation, 86 participants were diagnosed with mild dry eye, 29 with moderate, and 6 with severe forms. These findings suggest an increased risk of developing DESS in this population. Previous research using Shirmer I and BUT tests revealed varying degrees of DED severity, providing valuable insights into the prevalence of DED in healthcare workers and highlighting the need for effective prevention and management strategies. However, no studies to date have directly compared symptoms across different screen sizes, such as smartphones, computers, and tablets, or their effects on binocular function and tear film stability. Future research should also prioritize the use of emerging technologies to enable a more objective and accurate assessment of the tear film through imaging techniques, such as meibography and interferometry, for monitoring dry eye. In addition, wearable eye cameras and eye-tracking devices could facilitate continuous data recording during screen exposure [43].

The articles published on DESS and ocular surface anomalies present several common limitations. Firstly, although the CVS-Q questionnaire is widely translated and disseminated within both the scientific and clinical communities as a tool to identify DESS symptoms, only two studies employed this questionnaire. Secondly, while sample sizes in these studies were generally larger than those investigating binocular vision anomalies, they remained relatively small and not representative. A third frequently observed limitation was the self-reported screen time, rather than the use of objective measures.

## 4. Conclusions

This review identified substantial methodological heterogeneity among the included studies, most of which were cross-sectional, provided limited evidence on causality and failed to adjust for confounding factors. To advance the field, greater consistency in study design is needed to reduce sample bias, ensure population representativeness, and facilitate comparisons and replication across ethnically diverse groups. While randomized studies are already present in the field of dry eye disease, this is not yet the case in the field of binocular vision. Future research should implement rigorous protocols that account for confounding variables, employ RCTs, and prioritize the development of validated objective metrics for the diagnosis and management of DESS. Furthermore, longitudinal studies are necessary to enable the identification of whether the observed changes are pre-existing or if there are indeed patterns and shifts in clinical parameters that could characterize DESS as a distinct clinical entity. We found that one critical issue was the lack of blinding, which can introduce significant expectation and observer biases. Therefore, future studies should implement masking procedures to minimize these biases by concealing group assignments or specific participant information from researchers, assessors, or outcome evaluators. This approach enhances the validity and reliability of the findings. Another important point to consider is that refractive error should always be assessed to ensure studies are conducted with optimal optical correction. This helps eliminate bias associated with uncorrected refractive errors, which may cause asthenopic symptoms.

DESS presents with broad and overlapping clinical features, particularly with dry eye disease and binocular vision anomalies. It is essential to determine whether it represents a syndrome of overlapping visual and ocular surface stressors or a distinct clinical diagnosis, because although a direct causal relationship between digital device use and binocular vision changes remains unconfirmed, individuals with pre-existing binocular vision dysfunctions may experience DESS symptoms, such as blurred vision and headaches, increasing asthenopia, during prolonged near work. Understanding the pathophysiological mechanisms of DESS is especially critical in children, as accommodative and vergence systems are still developing and may be particularly vulnerable to the sustained visual demands imposed by extended digital device use. Clinical measures of binocular vision function, such as NPC, accommodative amplitude and facility, fusional vergence ranges, and phoria measurements using a cover test, alongside purely objective metrics obtained from automated systems (e.g., autorefraction-based accommodative response, video-based eye movement recordings, or eye-tracking measures such as fixation, blink rate, or viewing distance) during digital tasks, are crucial to accurately assess the effects of digital device use and to inform clinical management. Researchers at the University of Brunei Darussalam are developing the Digital Eye Strain and Risk Level Questionnaire (DESRIL-27) to provide a more comprehensive assessment of digital eye strain. Unlike previous tools, the DESRIL-27 includes two integrated scales: the Symptom Severity Scale, which measures symptoms related to the eyes, vision, and musculoskeletal system, and the Risk Level Scale, which evaluates various risk factors associated with the work environment. These include workspace setup, device or equipment use, work hours, break frequency, ergonomic practices, and other factors relevant to digital screen users [46]. Validation, especially for smartphone-specific issues, and minimizing recall bias are crucial for reliable assessment.

Given the global rise in screen exposure, longitudinal studies and RCTs are essential to elucidate the underlying pathophysiology of DESS and identify effective diagnostic tools and treatment strategies. Although pinpointing its precise mechanisms remains a challenge, DESS represents a growing public health concern due to its potential to impair visual performance and reduce quality of life.

## Figures and Tables

**Figure 1 jemr-18-00039-f001:**
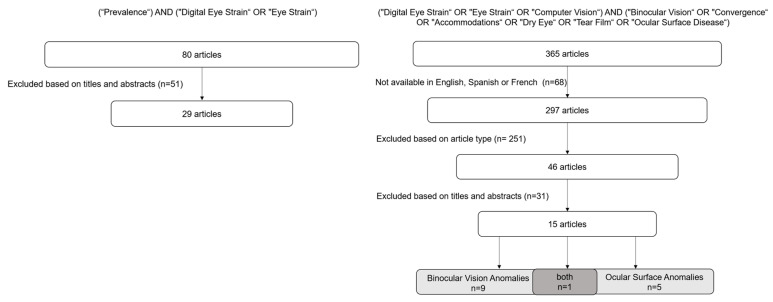
Flowchart of the stepwise search strategy and methodology.

**Figure 2 jemr-18-00039-f002:**
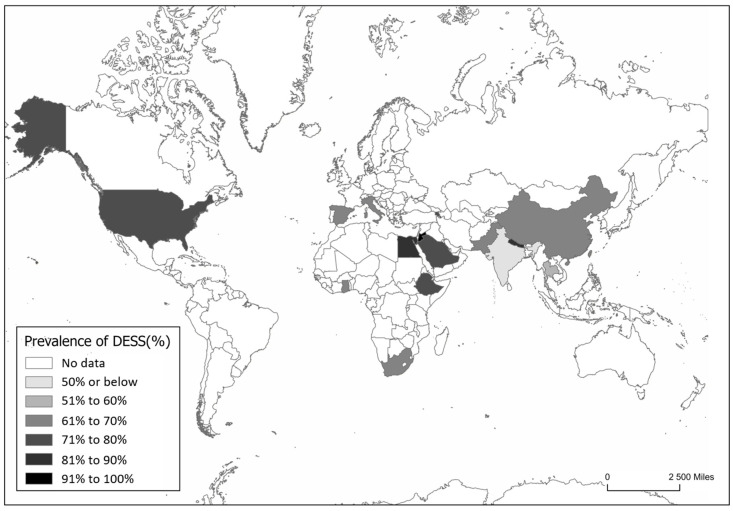
Worldwide prevalence of Digital Eye Strain Syndrome based on the weighted average of representative studies per country, as reported in 29 manuscripts published between 2020 and 2024. Countries shown in white represent those for which no prevalence studies on DESS were identified in the literature review.

**Figure 3 jemr-18-00039-f003:**
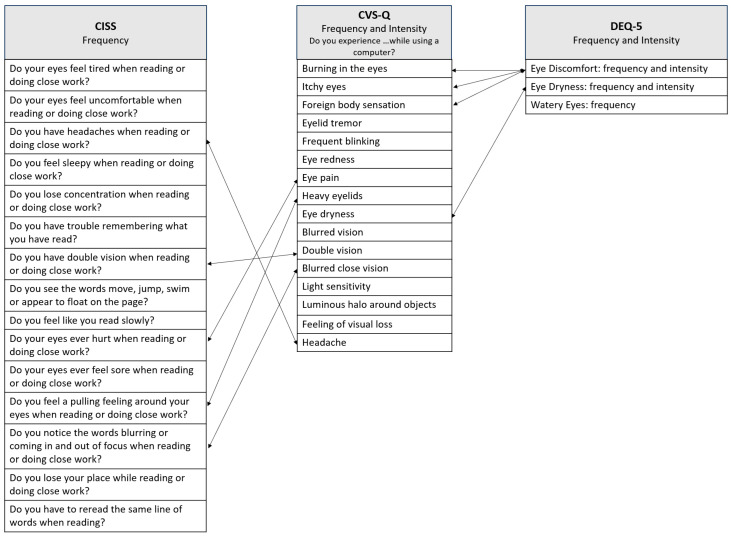
Relationship between the questions of the CVS-Q, CISS, and DEQ-5 questionnaires. CVS-Q—Computer Vision Syndrome Questionnaire; CISS—Convergence Insufficiency Symptom Survey; DEQ-5—Dry Eye Questionnaire version 5.

**Figure 4 jemr-18-00039-f004:**
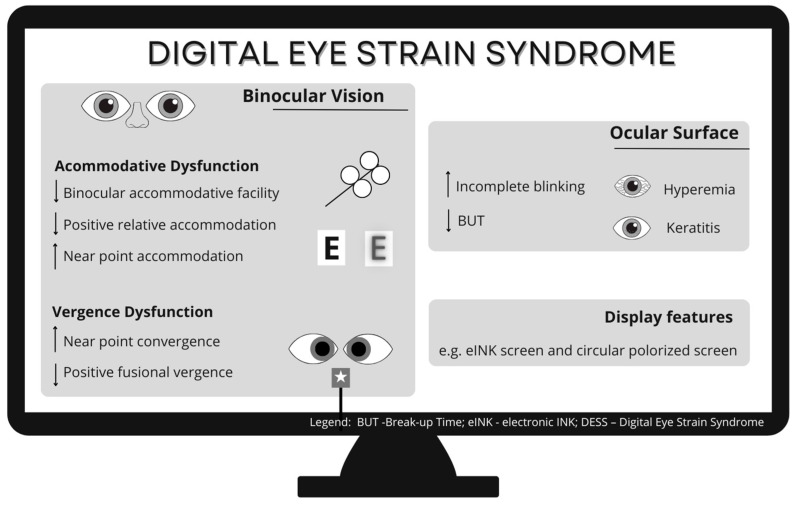
Potential contributing factors and mechanisms of Digital Eye Strain Syndrome. ↓ decrease; ↑ increase.

**Table 1 jemr-18-00039-t001:** Summary of Research Studies published on Digital Eye Strain Syndrome and Accommodative Dysfunction and Vergence Dysfunction.

Authors	Year	Country	Sample (*n*, Age, Years)	Type of Study	Methods	Questionnaire	Main Results	Limitations
Golebiowski, B., et al. [15]	2020	Australia	12 (18–23)	Cross-sectional (pilot study)	Evaluation before and after reading for 60 min on smartphone): ocular surface, horizontal fixation disparity and ease of binocular accommodation.	Questionnaires on eye strain and ocular surface symptoms.	↑ Symptoms after smartphone use (comfort, tiredness and drowsiness; *p* ≤ 0.02). Binocular accommodative facility (cycles/min) difference (CI 95%) = −3.0 (−5.7, −0.5), *p* = 0.01. Accommodative facility the median pre-task was 11.3 cycles/min (IQR 6.6), and the post-task was 7.8 cycles/min (IQR 2.5), *p* = 0.01.	Small sample size Short-term intervention
Sánchez-Brau, M., et al. [33]	2020	Spain	109 VDT workers mean = 54 ± 4.8	Cross-sectional	VDT exposure daily duration (hours), cumulative years of use, and screen technology. Viewing distance, eye-screen angle and neck posture. Frontocometer, refraction and binocular VA.	CVS-Q	↑ risk of CVS symptoms: Female: OR = 3.40; 95% CI, 1.12–10.33; *p* = 0.031. Non-neutral neck posture: OR = 3.27; 95% CI, 1.03–10.41; *p* = 0.045. Altered lighting: OR = 3.64; 95% CI, 1.22–10.81; *p* = 0.020.	Single Visit Self-reported screen time Limited assessment of lens characteristics
Yammouni, R., & Evans, B. [19]	2021	United Kingdom	107 mean = 31	Cross-sectional	Ophthalmoscopy, slit lamp biomicroscopy, retinoscopy, subjective refraction and accommodation and binocular tests.	CVS Q SANDE	Kendall’s tau correlation revealed no significant associations between amplitude of accommodation and % WRRT and CVS-Q. Other measures, such as accommodation lag, ±1.50 facility, NPC and vergence facility were also non-significant.	Lacks a control group without DES
De-Hita-Cantalejo, C., et al. [34]	2021	Spain	309 mean = 10.75 ± 0.67	Cross-sectional	Two groups: mild and severe CVS. Visual acuity testing, cover test, accommodation and vergence.	CVSS17	Breakdown (*p* = 0.03) and recovery (*p* = 0.02) of the NPC. Distance of break and recovery of NFV (*p* = 0.02 and *p* < 0.01, respectively).	Uneven distribution of symptom severity Lack of follow-up Limited generalizability due to sample setting
Liu, Z., et al. [35]	2022	China	93 mean = 24.09 ± 2.57	Cohort	Before and after a 20 min reading period: demographic characteristics and daily computer use. Eye movements and accommodation parameters.	CVS Q	CVS-Q score and PRA: *r* = 0.206 (*p* = 0.047); difficulty focusing for near vision was positively correlated with: TFF *r* = 0.279 (*p* = 0.007), TFD *r* = 0.235 (*p* = 0.023), TVD *r* = 0.253 (*p* = 0.014), and RS *r* = 0.237 (*p* = 0.022). Feeling of sight worsening was positively correlated with regressive saccades *r* = 0.27 (*p* = 0.011).	Small sample size Lack of ocular surface assessment Short-term intervention
Auffret, E., et al. [36]	2022	France	24 control mean = 28.6 ± 9.2 28 exposed mean = 35.2 ± 11.4	Cohort (pilot study)	Changes in binocularity in short-term exposure to the screen in 2 groups at 2 points in time: a group exposed <5/d and a group exposed >5 h/d. Assesses the consequences of chronic exposure—exposure to the screen >5 h/d, 5 days a week, for 1 year. Ocular discomfort questionnaire and binocular function.	Ocular discomfort questionnaire	Ocular discomfort score > in the exposed group: 0.3 vs. 0.6 (*p* = 0.04); blurred intermediate group control 0.3 and group exposed 1 (*p* = 0.02) vision; light sensitivity control group 0.8 and exposed group 1.4 (*p* = 0.04). FV in synoptophore—group control 18.09° and group exposed 13.42° (*p* = 0.045). BAF—group control 10.17 cycles/min and group exposed 6.00 cycles/min (*p* = 0.038). No significant differences were found between the control and exposed groups in NPC or NPA.	Small sample size Inclusion of participants with suboptimal optical correction
Wang, J. et al. [37]	2022	China	65 myopic mean = 20 ± 14.5	Cross-sectional	Subjective refraction, BCVA, surface ocular and binocular vision tests.	Questionnaire of asthenopia symptoms	57% of myopic patients had asthenopia; ↑ prevalence in older patients (*p* = 0.004). Asthenopia prevalence rate in myopic patients with <2 h of outdoor activities: 69%. Asthenopia prevalence in myopic patients with dry eye: 87%. Daily screen time (*p* = 0.003), continuous near work time (*p* < 0.001), eye care education (*p* = 0.002) and dry eye (*p* = 0.008) were positively correlated with asthenopia. Eye care education OR = 0.115 (*p* = 0.006) is a protective factor and continuous time working nearby OR = 4.227 (*p* = 0.046), indicating an increased risk of asthenopia.	Small sample size Non-standardized questionnaire
De-Hita-Cantalejo, C., et al. [38]	2022	Spain	118 (10–12)	Case–control	Two groups: low demand digital devices and high-demand digital devices. Visual acuity; accommodation amplitude, posture and facility.	CVS-Q	Only visual acuity showed a statistically significant difference between groups: visual acuity both eyes LDDD 1.22 ± 0.01 HDDD 0.62 ± 0.05 (*p* < 0.01).	Lack of follow-up Self-reported screen time Age group differences
Maharjan, U., et al. [21]	2022	Nepal	180 (7–17)	Cross-sectional	Ophthalmology and binocular vision examinations. Two groups: user group—digital devices in the last 6 months; non-user group—not used digital devices in the last 6 months. The user group was subdivided: low users (<3 h/d and 1 day a week) and high users (>3 h/d and every day of the week).	Parents were asked about the amount of time children use digital devices	Accommodative amplitudes, accommodative ease, and positive fusional vergence for both near and distance were significantly reduced in the high digital device user group (*p* < 0.01). ↑ Prevalence of accommodative and vergence anomalies (except vergence insufficiency) in the subgroup of high users of digital devices (*p* < 0.01).	Non-standardized questionnaire No assessment of subjective symptoms Focus solely on objective measurements
Cacho-Martínez, P., et al. [32]	2024	Spain	346 mean = 32.95 ± 14.56	Prospective study	Refractive examination: retinoscopy and subjective examination. Accommodative tests: Monocular AA, MAF and BAF (±2.00D), MEM dynamic retinoscopy, and PRA e NRA. Binocular vision tests: CT; PFV e NFV, Worth test and stereopsis.	SQVD	57.2% reported visual symptoms. 65.3%, some form of visual dysfunction (objective measure). <35 years, an association was found between having visual symptoms and digital device use (OR = 2.10, *p* = 0.01), adjusting for visual dysfunctions; this association disappeared (OR = 1.44, *p* = 0.27). >35 years, no association was found between symptoms and the use of digital devices (OR = 1.27, *p* = 0.49).	Self-reported screen time

Legend: DESS—Digital Eye Strain Syndrome; OSDI—Ocular Surface Disease Index; Cycles/Min—Cycle per Minute; CI—Confidence Interval; IQR—Interquartile Rage; VDT—Video Digital Terminal; CVS—Computer Visual Syndrome; CVS-Q—Computer Vision Syndrome Questionnaire; OR—Odds Ratio; SANDE—Symptom Assessment in Dry Eye; WRRT—Wilkins Rate of Reading Test; NPC—Near Point of Convergence; NFV—Negative Fusional Vergence; CVSS17—Computer Vision Symptom Scale 17; PRA—Positive Relative Accommodation; NRA—Negative Relative Accommodation; AA—Accommodation Amplitude; TFF—Time First Fixation; TFD—Total Fixation Duration; TVD—Total Visit Duration; RS—Reading Speed; FV—Fusional Vergence; BAF—Binocular accommodative facility; NPA—Near Point of Accommodation; BCVA—Best-Corrected Visual Acuity; LDDD—Low Demand Digital Devices; HDDD—High-Demand Digital Devices; D—Diopters; h/d—Hours per Day; MAF—Monocular Accommodative Facility; MEM—Monocular Estimate Method; CT—Cover Test, PFV—Positive Fusional Vergence; NFV—Negative Fusional Vergence; SQVD—Symptom Questionnaire for Visual Dysfunctions; ↑ increase.

**Table 2 jemr-18-00039-t002:** Summary of Research Studies Published on Digital Eye Strain Syndrome and Ocular Surface Anomalies.

Authors	Year	Country	Sample (*n* and Age, Years)	Type of Study	Methods	Questionnaire	Main Results	Limitations
Golebiowski, B., et al. ([15])	2020	Australia	12 (18–23)	Cross-sectional pilot study	Evaluation before and after exposure (reading for 60 min on smartphone) NIBUT, lipid layer appearance, TMH.	Questionnaires for eye fatigue and ocular surface symptoms.	↑ incomplete blinks per minute (median of 6 at baseline to 15 after 60 min of screen use; *p* = 0.0049). No significant differences were observed in TMH, NIBUT, lipid layer grade, or total blink rate over time.	Small sample size Non-standardized questionnaire Short-term intervention
Sanchez-Valerio, M., et al. [16]	2020	México	108 VDT workers mean = 32.1 ± 7.8	Cross-sectional online	Three groups: Mild Exposure Group (less than 3 h/d); Moderate Exposure Group (between 3 and 7 h/d); Severe Exposure Group (more than 8 h/d). TBUT, ocular surface staining and Schirmer test.	CVSS17 OSDI Questionnaire: time of exposure to the computer and the type of VDT used.	50.9% used a laptop, 38% desktop and 11.1% used both Average computer exposure time: 5.96 ± 2.5 h/d. 79.7% had symptoms of DED according to the OSDI. 97.2% had changes in tear breakup time, 44.4% had damage to the ocular surface and 26.9% had reduced watery tear production. Computer exposure time was positively correlated with DED and negatively correlated with TBUT (*p* < 0.001). Accumulated screen exposure time negatively correlated with TBUT (*p* < 0.001) and positively correlated with ocular surface damage (*p* < 0.001). No significant correlation was found between exposure time and Schirmer.	Self-reported screen time
Yuan, K., et al. [17]	2021	China	119 university students mean = 24.76	Prospective randomized controlled study	Continued reading for 2 h on different smartphone screens. Evaluation groups: Light + OLED; Light + eINK; Dark + OLED; Dark + eINK. Eye examinations: VA; NIBUT; lacrimal meniscus alterations; hyperemia; FBUT; CFS; Meibomius gland assessment, Shirmer I test; blink frequency. Parameters assessed BEFORE and AFTER 2 h reading tasks at 40 cm.	OSDI CVS-Q	iBUT after 2 h reading on the OLED screen compared to the baseline in light and dark environments. ↓ in tear meniscus after OLED reading (*p* < 0.001). ↑ blink rate with OLED screen (*p* < 0.001). ↑ocular hyperemia in the nasal and temporal area of the bulbar conjunctiva (*p* < 0.001). The eINK screen had a minor effect on tear film stability, tear volume and eye redness. Reading on an eINK screen did not exacerbate eye symptoms.	Short-term intervention
Mou, Y., et al. [18]	2022	China	120 university students mean = 25.86 ± 2.31	Prospective randomized controlled study	Evaluation groups with 30 participants: 4 groups: Circular + Dark; Circular + Light; Linear + Dark; Linear + Light. Eye examinations: CFF; TMH, NTBUT, redness of the conjunctiva, iBUT, CFS, Schimer I test.	OSDI CVS-17 CISS VAS	↑ OSDI scores of the linearly polarized light and dark (*p* < 0.001 and *p* < 0.001). ↓ NIBUT after reading in linearly polarized (*p* < 0.001). ↓ meniscus and Schirmer on linear. ↓ FTBUT in all groups, with the difference in ΔFTBUT present only between the linear and dark circular polarization (*p* < 0.05).	No adjustment for statistical multiplicity
Agarwal, R., et al. [4]	2022	India	435 mean = 35	Cross-sectional online	Online, questionnaire-based	Sociodemographic details, reason for increased screen time, number of hours, time spent on screens, common complaints and measures taken to overcome fatigue.	↑ screen time with lockdown (89% of participants); 81.4% had at least one symptom related to screens. Common symptoms were as follows: 52.8% eye pain/fatigue, 31.3% headache and 19.7% dry eyes/frequent blinking. 42.9% of participants reported that symptoms occurred frequently (daily or 2–3 times a week). 45.7% of participants adopted measures such as anti-reflective glasses, screen filters, increasing room lighting and decreasing screen brightness. Positive and significant correlation between the number and frequency of symptoms and the use of multiple screens, time spent on digital screens and continuous screen time.	Non-standardized questionnaire
Patel, H., et al. [42]	2023	India	501 mean = 23.14 ± 2.47	Cross-sectional study	Visual acuity using and anterior segment examination with a slit lamp. Patients with a score of 6 or more underwent dry eye examination tests: TMH, TBUT, and the Schirmer I test.	CVS-Q; questionnaire on awareness of CVS and use of digital gadgets. Questionnaire assessing the frequency and intensity of symptoms over time. Questionnaire on knowledge, attitude and prevalence of dry eye.	Most common symptom was headache, 66.7% occasionally and 6.8% always. 47.7% reported that closing their eyes provided relief from dry eye. 56.1% used mobile phones and laptops. 46.3% reported using digital devices for 3–5 h and 35.7% using them for 1–3 h.	Self-reported screen time

Legend: DESS—Digital Eye Strain Syndrome; VDT—Video Digital Terminal; BCVA—Best-Corrected Visual Acuity; TBUT—Tear Break-Up Time; CVSS17—Computer Vision Symptom Scale 17; OR—Odds Ratio; CI—Confidence Interval; OSDI—Ocular Surface Disease Index; mm—Millimeters; NIBUT—Non-Invasive Tear Break-up Time; iBUT—Invasive Tear Break-up Time; DED—Dry Eye Disease; OLED—Organic Light-Emitting Diode; eINK—Electronic INK; VA—Visual Acuity; CISS—Convergence Insufficiency Symptom Survey; VAS—Visual Analogue Scale; CFF—Critical Flicker Fusion Frequency; FTBUT—Fluorescein Tear Break-up Time; ΔFTBUT—Range of Fluorescein Tear Break-up Time; CFS—Corneal Fluorescein Staining; CVS—Computer Visual Syndrome; TMH—Tear Meniscus Height; h/d—Hours per Day; ↑ increase; ↓ decrease.

## Data Availability

The original contributions presented in this study are included in the article/Appendix A. Further inquiries can be directed to the corresponding authors.

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
