# Peer review of "A Review of Digital Eye Strain: Binocular Vision Anomalies, Ocular Surface Changes, and the Need for Objective Assessment"

_1995-8692, 2025, doi:10.3390/jemr18050039_

Round 1
Reviewer 1 Report
Comments and Suggestions for Authors
I think that this review paper aims to summarize observational in the literature that examined the relationship between digital eye strain and binocular outcome measures and digital eye strain and dry eye. Of the 32 observational studies identified, 16 were related to accommodative or vergence dysfunctions and 8 were related to ocular signs and symptoms of dry eye. The authors begin the presentation of their findings with the prevalence of digital eye strain across the world based on the 32 studies though it is not a primary outcome. Then they present the binocular outcomes, which is their primary goal, and lastly they present the ocular surface outcomes which is their secondary goal.
I had a very hard time following the logical flow and think the paper needs extensive rewriting ,as I will describe below.
- What is the justification for this review. Please state previous reviews, you can limit the reviews to post-COVID as both display types have changed and post-COVID work habits have changed. Have previous reviews examined the topic of binocular outcomes related to digital eye strain? If they have- what is the benefit of the present review? The same question applies to the ocular surface parameters- have previous reviews examined the topic of ocular surface outcomes related to digital eye strain? If they have- what is the benefit of the present review?
The authors included refractive error under ocular surface parameter. But it is a different characteristic, and should be dealt with separately - What type of review is this ? The abstract says narrative review, but this is not mentioned in the body of the manuscript
- Logical flow I- to facilitate reading of the manuscript, a flow chart is recommended. Show us how many papers were detected, how many passed the initial screening due to exclusions, how many were related to vergence/accommodation and how many were related to ocular surface parameters and how many to refractive errors
- Logical flow II- If the primary goal is to discuss the relationship between digital eye strain and binocular outcomes, followed by digital eye strain and ocular surface outcomes followed by digital eye strain and refraction- the order of presentation should follow the logic. The prevalence was not a primary goal and should not appear first.
- Logical flow and justification. In the introduction, the authors mention one study
A small scale cross-sectional study involving young adults investigated the impact of 60 minutes of smartphone use on binocular vision and found a post-task reduction in accommodative facility (11.3 cycles/min vs 7.8 cycles/min) (11).
-This finding is controversial as several other studies do not find impacts on accommodation. Why was just one study selected? Perhaps this is a good place to provide studies that do show impact on accommodation and some that do not, thereby providing additional justification for the study. In the results, the authors should devote a section that tries to resolve this discrepancy and provide potential reasons for the conflicting findings. - Unclear methods- Were only randomized control trials included or any studies that included binocular outcomes and digital eye strain/ ocular parameters and digital eye strain/ refraction and digital eye strain? What are the exact exclusion and inclusion criteria for the selected papers? For example, Study 33 did not assess digital eye strain at all. Why was it included?
- Logical flow in outcomes. This is a review which summarizes previous studies and is supposed to contrast and compare between them highlighting points of similarity and discrepancy. One important point that the authors make is the diagnosis of digital eye strain DESS in the papers included. Each one is based on a subjective questionnaire, and the questionnaires differ. The authors should highlight this in a dedicated section and provide numbers that show how many of the included studies (13 in the vergence/accommodation section) based the diagnosis on the CVS questionnaire and how many were based on other questionnaires? Also provide some thought and discussion to which questionnaire is more reliable and valid for the diagnosis.
- Unclear logic and discussion lines 166-168 : "In addition, current questionnaires focus on computer-related symptoms, overlooking challenges from shorter distances and smaller screens, such as smartphones, which strain accommodation and convergence."
Based on what do the authors claim this? Most questionnaires ask about reading without specifying the distance. Therefore I'm not clear as to why the authors are suggesting that the current questionnaires are not applicable to shorter working distances? - Language: Overall the manuscript's English is clear. However there were some areas that required modifications due to incorrect grammar and wording as described below.
Minor comments and edits
Lines 60-61 Digital devices differ from paper-based reading, potentially disrupting the visual system, especially binocular vision, though the extent and nature of this effect are not clear.
-Provide a reference
Lines 64-66: In addition, university students and digital professionals, often use multiple devices for long periods. In university settings, most students currently rely on electronic study methods, with approximately one-third reporting frequent use of virtual learning
- provide a reference
Figure 1 - what is lack of eligible data? Was the prevalence not reported? Or did the paper not meet eligibility criteria for inclusion in the review?
"CI is characterized by near exophoria, a remote near point of convergence, and reduced positive fusional vergence".
- Typically the first instance of abbreviations should be spelled out. The authors did this in other sections of the manuscript. A list of abbreviations was added at the end. Not sure what MDPI guidelines are . Also given that the same abbreviation in this manuscript refers to two different concepts, perhaps it is just better to spell it out?
- Additionally, a reference is missing for the statement
Regarding Reference 31, the authors state : "However, no significant differences were found in near point convergence or near point accommodation (NPA) between high-exposure (5 h/d) and control groups (less than 5 h/d) (31)."
- How many participants? This is warranted granted the authors specifically state that their findings are limited by the sample size
Line 198: "Moreover, although the study compared refractive error values obtained via autorefractor measurements with those present in the participants’ current spectacles, the absence of updated refractive prescriptions where clinically indicated constitutes an additional methodological limitation."
- If the study team verified using an autorefractor that the current prescription doesn't deviate much from autorefraction why do the review authors think that this is a study limitation? Was it an exclusion criteria when the deviation between the existing spectacles and the auto refraction was large? What was the mean difference and median difference between the two? I assume that we're talking about differences of half diopter or less.. if the differences are larger than the review authors have a good point but the numbers should be provided..
-Line 203, delete the word moreover
-Line 209 shorter "visit" duration or reading duration?
-Lines 213-214
Please State how many participants were in that study otherwise stating that a larger sample size is necessary is completely unfounded
- Line 215 - the word "although" seems inappropriate here.
Rephrase. Perhaps "unlike the study focusing on short durations of reading, the Portuguese Group of Ergophthalmology assessed the effects of daily screen time longitudinally over a one-month period"
- Lines 220-221- office workers wore screens or "used" them ?
-Lines 226-227- provide a reference
- Line 239: "the lack of robust evidence directly linking visual anomalies to DESS,"
Is this statement relevant only to this particular study. Doesn't it apply to all the studies stated thus far and in general?
-Similarly, in Line 251: "a notable limitation is the absence of a gold standard for diagnosing binocular vision disorders" , doesn't this limitation apply to all the studies described and in general?
- Existing studies often suffer from design flaws, small sample sizes, non-representative populations, etc.- for each one specify how many of the studies that you included suffer from each one of the items or even add the specific limitations to the summary table.
- Line 321: how much was the small sample size? Why was the sample not representative?
- Tear film section devotes a major part of the discussion to the effects of display characteristics on ocular surface, perhaps it may be better to describe this in a separate section with an appropriate header?
- please describe statistical mutliplicity and why it causes a lack of clinical significance
- Line 381 the same research group provided conflicting findings about the relationship of blinking and digital display use. Can you provide possible reasons for the discrepancy?
- Line 482 and refractive errors.
- Not included in the review but could be very useful : what are these "objective" measures of binocular outcomes and digital display use that the authors are recommending
- Line 495 interdisciplinary research involving which additional professions? Why? This conclusion is not supported by the data presented and just came from no where
See comment 9
Author Response
We would like to express our sincere gratitude for the thorough review of our article. The comments and suggestions were important to enhance the quality of our manuscript. We have carefully incorporated these suggestions into the manuscript, which are highlighted using the 'track changes' feature.
Our response an attachment.

Reviewer 2 Report
Comments and Suggestions for Authors
A Review of Digital Eye Strain: Binocular Vision Anomalies, 2 Ocular Surface Changes, and the Need for Objective Assess-3 ment
The abstract is well written. It summarizes the current state of digital eye strain diagnosis and the limitations of subjective analyzes that exist in the literature today.
The authors divide the studies into those looking at binocular abnormalities, dry eye impacts and refractive error impact on DESS. This does make the review a little more difficult to follow, but more comprehensive.
I believe that more strabismus terms should have been used in the search for articles regarding binocular dysfunction, such as “exotropia, esotropia, strabismus, diplopia, asthenopia” to try to capture binocular dysfunction and digital usage articles.
I am still unclear what subjective/objective finding is indicative of DESS. It seems that any combination of dry eye, convergence or accommodation dysfunction in the presence of someone who is using a digital device for prolonged periods is given the diagnosis of DESS. If this is not a separate entity, but rather an exacerbation of symptoms due to dry eye, convergence or accommodation dysfunction then it is not a separate diagnosis and treatment would need to be focused on the underlying disorder. Lines 74-80: The authors imply this in their introduction, but then state that more research is needed. Lines 84-87: The work also explores dry eye syndrome and refractive error, so not only focused on binocular dysfunction.
I think the introduction needs a better description of the type of binocular dysfunction that is studied in DESS. Define positive relative accommodative, negative relative accommodation, and any other terms. Qualify which measures are most important. How was accommodative facility measured—objective blur, # of times flipper rotated in set time? How was NPA measured in studies—Prince (RAF) ruler or other methods? This is the background needed to assess the studies done. Also why is convergence insufficiency different than DESS?
Similarly, perhaps add to introduction the differences between pre-existing dry eye syndrome and worsening with reduced blinking while using the computer compared to ocular surface symptoms that exist in DESS exclusively due to computer usage. How do we differentiate the two?
In regards to prevalence, I believe that you can only state that DESS has been reported in the countries where articles have been published, but the converse that if there are no articles, implies that DESS is less common cannot be proven. The results should reflect that this is the map of published studies only and the impacts in other countries remain unknown—not that DESS doesn’t occur in those countries.
Figure 3. I believe the spelling is dysfunction not disfunction.
Why is article reference 26 not included in the meta-analysis?
Line 183: Please clarify that the articles were ranked as moderate on the Newcastle- Ottawa scale.
Lines 184-193. Since this is a review of these articles, I think it would be important to point out if finding a positive correlation in accommodative anomalies with high computer usage is a change in the individual’s accommodation, or was poor accommodation pre-existing in the individuals tested? If patients were not wearing their appropriate refractive correction, then the amount of accommodation would vary depending on the extent of under/overcorrection of their refractive error. Also how many patients were included in the study sited—ref 31.
Line 220: what does it mean to “wore screens”—viewed screens or some type of virtual reality use?
Why does the font change from lines 337-344?
I suggest organizing the 16 binocular impact studies into those that found accommodative dysfunction compared to vergence dysfunction. Currently, there seems to be a commentary on each in no apparent order. Those studies with strongest evidence should be highlighted. If patients with pre-existing binocular problems were included within the group of computer users, then we don’t know if it is computer usage or some other factor resulted in greater percentage of binocular problems. This limitation needs to be highlighted.
Line 433 should be “refractive error”, not “error refractive”
Line 446 what is ophthalmologic education?
Lines 455-456: In addition, overminused refractive correction would increase accommodative demand while participating in near work.
The conclusion is well written. I think this paper does a good job of summarizing the state of research on DESS. It points out inconsistency in the diagnosis, and biases in studies on impact of device usage when not blinded.

Author Response

(The authors gave the same response as above.)

Reviewer 3 Report
Comments and Suggestions for Authors
The authors present in this manuscript a narrative review of Digital Eye Strain Syndrome (DESS), with a particular emphasis on binocular vision dysfunctions and the need for objective diagnostic approaches.
- I suggest elaborating on how digital reading disrupts the visual system compared to paper-based reading to clarify the physiological mechanisms involved.
- I suggest clarifying whether the 32 prevalence studies overlap with the 16 main articles or are entirely distinct.
- Could you clarify the specific inclusion and exclusion criteria applied when selecting studies for this review?
- Could the review clarify why the included studies were limited to publications in only three languages?
- Given the increasing reliance on smartphones and tablets, have you considered the need for a new or adapted questionnaire specifically targeting symptoms from shorter viewing distances and smaller screens?
- To provide a more comprehensive understanding of the impact of digital device use on the ocular surface, it would be valuable to include recent studies investigating the effects of blue light exposure and the long-term consequences of continuous screen use.
Author Response

(The authors gave the same response as above.)

Round 2
Reviewer 1 Report
Comments and Suggestions for Authors
The authors have addressed the majority of my previous comments and suggestions. However I do have some comments and reservations on the additions. Further, I still did not see justification for the refraction and I am confused as to why the studies from 2017 are included given that the authors claim that they only included post COVID papers.
Comment 1- Lines 60-52: require references Digital reading devices differ from traditional paper-based media and are believed to influence the visual system, particularly binocular vision.
Comment 2- Line 70- what are digital professionals?
Comment 3- Line 87- longer screen time instead of higher
Comment 4- Line 88- Additionally instead of In addition
Comment 5- Line 90 should be a new paragraph and should come after the sentence in line 104 “Healthy binocular vision”. It is very confusing that the authors discuss binocular issues and DESS and then ocular issues and DESS and then revert back to binocular issues
Comment 6- Line 96: To distinguish- should be a new paragraph and begin with “Furthermore,”
Comment 7- No justification is provided for refraction and DESS like is provided for binocular issues and ocular issues
Comment 7- Materials and Methods- if the review is post-COVID, then studies from 2017 cannot be included, the lockdown was in 2020
Comment 8- Line 140- all the studies included used a RAF ruler? Not sure about the methodology of the clinical test measurements stated here
Comment 9- Figure 2 caption- left square bracket seems to be backwards
Comment 10- Line 187- seems multifactorial or IS multifactorial?
Comment 11- Line 271: accommodation amplitudes and facility,
Comment 12- Line 309: perhaps “other” near vergence assessments? Because fusional vergence amplitudes were measured..?
Comment 13- Line 320-321: While the study reported the proportion of participants considered to have appropriate optical correction, it did not provide data on those exceeding this threshold.- this makes no sense. If the proportion of participants with appropriate optical correction is provided, and the number of participants in the study is provided- then the proportion of those without the appropriate optical correction can be calculated,
Comment 14- Lines 314-324: Generally this paragraph is very wordy and should be reworded clearly and concisely.
Comment 15- Line 325: should be a new paragraph
Comment 16- Line 331: The Nepalese study- which one is this?
Comment 17- Line 346: should be a separate paragraph
Comment 18- 603-609: “Objective measurements of binocular vision function, such as NPC, accommodative amplitude, accommodative facility, fusional vergence ranges, and phoria measurements using prism cover test or Maddox rod- these are all not objective measures as they require patient responses. If the authors had described a cover test for measuring phoria (which is also not really objective), that would have been at least partially justified. In any case NPC, amplitude of accommodation, facility and ranges all depend on user responses to reach their measurable end points and therefore not at all objective.
Reviewer 3 Report
Comments and Suggestions for Authors
The authors have responded to my suggestions, and I have no more questions.
Author Response
Comment 1: The authors have responded to my suggestions, and I have no more questions.
Response 1: We are grateful for the comments and suggestions provided during the review process, which have helped us to improve our manuscript.
Round 3
Reviewer 1 Report
Comments and Suggestions for Authors
The authors addressed previous reviewer comments and suggestions. The manuscript is clearer now. However, there are some issues that should be further addressed:
Line 141 measures in all included studies using a RAF ruler
Line 222 Near tasks instead of close-up
Lines 235-245 should be reworded and a reference number should be added for Aufret et al.:
Similarly, in a cohort study conducted amongst university hospital employees in France, fusional amplitude (tested using a synoptophore) and binocular accommodative facility were reduced amongst 24 employees with more than 5 h/d of screen use compared to 28 employees with less than 5 h/d of screen use (Aufret et al. [missing reference number], Table 1). Near Point of accommodation was did not differ between the groups. The study did not specify if the refractive correction was verified or standardized prior to testing. Thus, accommodative perturbations (due to under or over-correction) cannot be ruled out and could introduce bias in the measurements.
Was dry eye evaluated here? Because it was stated as a limitation for the study described in line 256, also were symptoms assessed? Because they were only stated as limitation for the study in 267
Line 266: reinforcing the association between extensive screen use and impaired accommodative function
Line 288: a subsequent study instead of another study
Lines 300-310 discuss the same Auffret study as lines 235-245- only one instance should appear
Line 319: In a younger population from Spain, vergence disorders were more prevalent amongst children with severe DESS compared with mild DESS (38).
Lines 426-427: Reduced blink rate and incomplete blinking have been identified as key contributors to DESS, particularly during prolonged smartphone use.- missing a reference
Line 432: consider replacing the word “underscore” with “highlight” – this word is very commonly used throughout the manuscript
Line 465- In contrast, another study
Line 479-480: replace underscore with another word
What are the other “preventative measures” that should be adopted?
Lines 484-487 missing references:
These two types of DED are related with risk factors such as older age [[here]] elevated stress levels [[here]], and poor self-perception of health [[here]]. Aqueous deficiency dry eye was associated with being female and reduced sleep [[here]], while evaporative dry eye was linked to East and South Asian ethnicity [[here]], contact lens use [[here]], and increased screen time [[here]].
Lines 488- 489: what are the criteria for sample selection and modifying risk factors for DED prevention and treatment?
Line 497: is this related to the study described in the preceding paragraph ?
Line 505: wear eye cameras and eye trackers for a full day of tracking? Not clear
Line 523: remove longitudinal since you elaborate on it in the subsequent sentence
Comments on the Quality of English LanguageI would recommend having an academic editor or a person who is a native English speaker and academic read through the paper to correct its writing and improve its flow.
